# Characterization of A-to-I Editing in Pigs under a Long-Term High-Energy Diet

**DOI:** 10.3390/ijms24097921

**Published:** 2023-04-27

**Authors:** Liu Yang, Lei Huang, Yulian Mu, Kui Li

**Affiliations:** 1Shenzhen Branch, Guangdong Laboratory for Lingnan Modern Agriculture, Key Laboratory of Livestock and Poultry Multi-Omics of MARA, Agricultural Genomics Institute at Shenzhen, Chinese Academy of Agricultural Sciences, Shenzhen 518124, China; 2State Key Laboratory of Animal Nutrition and Key Laboratory of Animal Genetics, Breeding and Reproduction of Ministry of Agriculture and Rural Affairs of China, Institute of Animal Sciences, Chinese Academy of Agricultural Sciences, Beijing 100193, China

**Keywords:** pig, high-energy diet-induced, A-to-I editing, gene regulation

## Abstract

Long-term high-energy intake has detrimental effects on pig health and elevates the risk of metabolic disease. RNA editing modifying RNA bases in a post-transcriptional process has been extensively studied for model animals. However, less evidence is available that RNA editing plays a role in the development of metabolic disorders. Here, we profiled the A-to-I editing in three tissues and six gut segments and characterized the functional aspect of editing sites in model pigs for metabolic disorders. We detected 64,367 non-redundant A-to-I editing sites across the pig genome, and 20.1% correlated with their located genes’ expression. The largest number of A-to-I sites was found in the abdominal aorta with the highest editing levels. The significant difference in editing levels between high-energy induced and control pigs was detected in the abdominal aorta, testis, duodenum, ileum, colon, and cecum. We next focused on 6041 functional A-to-I sites that detected differences or specificity between treatments. We found functional A-to-I sites specifically involved in a tissue-specific manner. Two of them, located in gene *SLA-DQB1* and near gene *B4GALT5* were found to be shared by three tissues and six gut segments. Although we did not find them enriched in each of the gene features, in correlation analysis, we noticed that functional A-to-I sites were significantly enriched in gene 3′-UTRs. This result indicates, in general, A-to-I editing has the largest potential in the regulation of gene expression through changing the 3′-UTRs’ sequence, which is functionally involved in pigs under a long-term high-energy diet. Our work provides valuable knowledge of A-to-I editing sites functionally involved in the development of the metabolic disorder.

## 1. Introduction

The metabolism process breaking down proteins, carbohydrates, and fats and absorbing them into the body, is a process of energy transfer, absorption, and storage [1]. When abnormal chemistry reactions disrupt the normal metabolic process, metabolic disorders can happen [2]. To simulate these abnormal reactions, a pig model for metabolic syndrome was built by long-term high-energy diet-inducing in our previous work [3]. We also previously analyzed their metabolic parameters, atherosclerotic lesions, gut microbiome, and host transcriptome and revealed the correlation of gut microbiota with metabolic parameters and atherothrombosis [3,4].

RNA editing modifying RNA base is a post-transcriptional process [5,6]. Adenosine deaminase acting on the RNA (ADARs) enzyme mediates adenosine-to-inosine (A-to-I) editing, turning adenosine (A) to inosine (I) recognized as guanosine (G) [7]. A-to-I editing is the most widespread form of RNA editing in mammals. Many functions of A-to-I editing have been reported previously, such as modifying the sequence of RNA precursors [5,6], changing mRNA isoforms, recoding proteins, and resulting in physiological and pathological changes [8,9,10]. A-to-I editing events often occur in the non-coding regions, and they are significantly enriched in 3′ untranslated regions (3′-UTRs) transcripts [11] and play an important role in RNA degradation and nuclear retention [12,13,14]. Many of the A-to-I editing sites were identified in non-model animals, such as cattle [15], goat [16,17], sheep [18], chicken [19], and pig [20,21,22,23], while their functional characteristic has not been studied well. Several factors, such as temperature, tissue type, genotype, diet, and age, can influence RNA editing [19,24]. The study used RNA sequencing data to identify A-to-I editing sites in the pig genome and found that A-to-I editing was widespread in pigs, occurring in over 10,000 genes. The study also found that A-to-I editing levels varied between different tissues and developmental stages, suggesting that A-to-I editing may play tissue-specific roles in pigs [25].

The pig is an ideal animal model for human diseases because it has a closer genome size than the mouse genome that the human implying pig has more similar genes and their functions to humans [26]. Pigs also have anatomical and physiological similarities with humans. To study human metabolic disorders, the pig is a better choice than small rodent mice or rats whose small size leads to difficulties in performing surgery and using organs. In addition, metabolic rate is influenced by body size. The closer body size can produce more similar metabolic parameters that accurately estimate disease stages [27,28,29]. Not only that, pigs are a valuable source of protein, as their meat is a staple food item for people. The pork industry is a major contributor to the global economy. Maintaining good pig health is essential for the sustainability and profitability of the pig industry, as well as for ensuring food safety and animal welfare.

A-to-I editing is a valuable resource in the gene regulation process; however, little is known about its roles in pig health and metabolic disorders. In this study, to explore the functional characteristic of A-to-I editing in pig tissues under a long-term high-energy diet, we systematically analyzed the A-to-I editing profiling in three tissues and six gut segments, compared the difference of editing sites between high-energy diet treated and normal chow diet control pigs, characterized the functional aspect of editing sites in gene regions. This study might provide light on A-to-I editing sites functionally involved in the development of the metabolic disorder.

## 2. Results

### 2.1. A-to-I Editing across Pig Genome

Our study began with a total of 72 samples from eight individuals and encompassed nine tissues, including duodenum (Du), jejunum (Je), ileum (Il), colon (Co), cecum (Ce), rectum (Re), abdominal aorta (AA), liver (Li), and testis (Te). The samples were sequenced using strand-specific, rRNA-free, and pair-end platforms (Appendix A). RNA editing analysis was performed using REDItools2, and all types of RNA editing were detected. Sites were filtered based on known SNPs, individual genome variations, fewer than three samples, and 100% editing levels in all detected samples, resulting in 1,011,050 (average (ave.) 14,042.4) high-quality autosomal RNA eating sites across 72 samples. These sites were found to be located in 94,259 non-redundant positions in the pig genome. Among them, 64,367 (68.3%) corresponded to AG modification or A-to-I editing, with an average of 8962.5 for each of 72 (645,301 for all) samples (Figure 1a and Appendix A). The editing level of each site was defined as the proportion of editing reads in the total sequenced reads and was used to quantify the editing dosage for each site. The editing level of the editing sites for each type is presented in Appendix A. The majority of these sites (62.8%) were unevenly distributed in the middle of chromosomes, as shown in Appendix A. In all 72 samples, A-to-I editing sites were found to have an average editing level of 0.33, with a clustering tendency that was most prevalent within 20 base pairs of each other (Appendix A). Over 60,000 A-to-I editing sites were detected across all 72 samples, with the editing level peaking at around 0.25, as presented in Appendix A. The frequency distribution of A-to-I sites, depicted in Appendix A, revealed that the majority of sites (17.7%) were detected in only three samples. To explore the functional aspects of A-to-I editing sites in pigs, we annotated 64,367 non-redundant sites against gene regions using the Ensembl gene annotation file. Among these sites, 55,986 (87.0%) were located in annotated regions, with enrichment observed in protein-coding genes and regions transcribing RNAs that regulate gene function, such as miRNA, miscRNA, scaRNAs, and snoRNA (Figure 1b and Appendix A). Of the 55,986 A-to-I editing sites in the annotated region, 96.2% were located in 14,226 transcripts of 5251 PCGs. These sites were distributed across different gene features, including exons, start and stop codons, 5′-UTRs, CDSs, and 3′-UTRs (Appendix A). A-to-I editing sites were significantly enriched in 5′-UTR and 3′-UTR, whereas sites in CDS were significantly lower than the expectation when compared with the total length of gene features in the pig genome (Figure 1c). These findings suggest that A-to-I editing plays a role in regulating gene function.

### 2.2. Functional Aspects of A-to-I Editing across Pig Tissues

To investigate A-to-I editing characteristics across pig tissues, we compared editing patterns among six gut segments (Du, Je, Il, Co, Ce, and Re) (Figure 1d,e), as well as three tissues from important organs (AA, Li, and Te) (Figure 1f,g). We found that the colon had the highest A-to-I editing frequency (ave. 8755.1) among the six gut segments, followed by cecum (7653.4), ileum (5944.5), rectum (5900.9), duodenum (5398.5), and jejunum (5130.5). The rectum had the highest average editing level (0.313) for all sites, followed by the colon (0.310), cecum (0.309), ileum (0.309), duodenum (0.306), and jejunum (0.300). Additionally, the abdominal aorta exhibited a unique A-to-I editing profile, with the highest editing frequency (ave. 27927.2) and the highest editing level (ave. 0.371) among all tissues. In contrast, the liver had the lowest editing level for all sites (ave. 0.296) and an average of 8051.2 A-to-I editing sites. The testis had intermediate levels of editing site count (ave. 5901.4) and editing level (ave. 0.314). We next performed Upset plots to illustrate A-to-I editing overlaps across tissues. As expected, it showed that the abdominal aorta processed the largest number of specific A-to-I editing sites (32,419 non-redundant) than the liver (5874) and testis (3032), and three tissues shared 7457 sites (Appendix A). For these six gut segments, a maximum number of A-to-I sites (7312) were shared by six tissues, specific editing site in the cecum was the highest (2272) followed by colon (1221), rectum (1220), duodenum (961), jejunum (457) and ileum (452) (Appendix A).

### 2.3. Changes in A-to-I Editing for the Long-Term High-Energy Diet of Pigs

As described previously, we treated half of the pig individuals with a long-term high-energy diet (HED, with added 10% fat, 3% cholesterol, and 87% base material) for 64 months, starting with pigs aged 3 to 4 months. The non-redundant A-to-I editing sites for HED and normal chow diet (ND, with 20% wheat, 48% corn, 15% soybean cake, 5% fish meal, and 12% rice bran) treatment of each tissue of pigs were shown in Table 1. To systematically investigate the effects of long-term high-energy intakes on the A-to-I editing pattern in different tissues, we compared the difference between HED and ND groups based on 64,367 non-redundant A-to-I sites. Principal component analysis (PCA) was conducted on the editing levels of each site for 72 samples, which separated the tissues but did not reveal a clear distinction between HED and ND pigs (Figure 2a). The count of A-to-I editing sites for each tissue between the HED and ND groups was also not significantly different (Wilcox Test, *p* > 0.05) (Figure 2b,c). However, there was a significant difference in editing levels between the two groups for the abdominal aorta, testis, duodenum, ileum, colon, and cecum (Wilcox Test, *p* < 0.05). The testis exhibited the largest difference, with an average increase of 5.36% (0.016) in editing level for all sites after 64 months of HED treatment, followed by a decrease in editing levels in the duodenum by 3.50% (−0.011), ileum by 3.23% (−0.010), and an increase in editing levels in the colon by 2.17% (0.007), cecum by 2.03% (0.006), and abdominal aorta by 0.68% (0.003). These results suggest that a long-term high-energy diet primarily increases editing levels in the testis more than in the abdominal aorta and liver while mostly decreasing editing levels in the duodenum among the six gut segments.

We conducted pair-wise comparisons between HED and ND pigs for each of the nine tissues using the Wilcox Test method. We focused on 19,629 A-to-I editing sites that were represented in at least three samples for each treatment of tissue and found that 433 sites (2.2%) showed a significant difference (*p* < 0.05) between HED and ND across tissues (Figure 2d). Among these sites, 278 (64.2%) showed increased editing levels after HED treatment, while 155 (35.8%) showed decreased levels. For A-to-I editing sites that were only detected in one treatment of tissue and not in the other treatment samples of the same tissue, we defined them as treatment-specific editing sites of that tissue. Across the nine tissues, we detected a total of 3550 HED-specific and 2191 ND-specific editing sites (Appendix A and Table 2), making up 9.4% (6041/64,367) of the functional A-to-I editing sites detected as being different or specific to HED or ND treatment across tissues. The highest count of such sites was found in the abdominal aorta (4.2%, 2027/47,931), while the highest proportion was in the liver (4.7%, 823/17,375) (Figure 2e).

### 2.4. Gene Enrichment of Functional A-to-I Editing Sites

#### 2.4.1. Functional A-to-I Editing Sites Involved in Aspects of the Living Process

To analyze the functional implications of the 6041 functional A-to-I editing sites, the overlaps between them were compared. The majority of the 3075 functional sites observed in the six gut segments were found in the cecum (23.3%, 715), which was also the case for the abdominal aorta, which has the most A-to-I editing sites (Figure 2f and Appendix A). Furthermore, 2206 protein-coding genes were annotated by 4998 functional A-to-I editing sites across all nine tissues, and functional enrichment analysis was performed (Appendix A). This analysis revealed that the functional A-to-I editing sites were associated with 790 gene ontology terms related to diverse biological processes such as adhesion, apoptotic, catabolism, and modification. Examples of enriched GO terms included cell-substrate adhesion, regulation of extrinsic apoptotic signaling pathway, positive regulation of the catabolic process, regulation of protein catabolic process, peptidyl-serine phosphorylation, protein polyubiquitination, negative regulation of extrinsic apoptotic signaling pathway, cellular response to peptide hormone stimulus, regulation of apoptotic signaling pathway, the establishment of organelle localization, and response to transforming growth factor beta (Appendix A).

#### 2.4.2. Functional A-to-I Editing Sites Involved in Tissue’s Specific Function

Each tissue exhibited unique functions enriched by its functional A-to-I editing sites. For example, the abdominal aorta had 1766 functional sites located in 952 genes enriched in cell-substrate adhesion, focal adhesion assembly, Wnt signaling pathway, and hormone-mediated signaling pathway. The liver had 627 sites that significantly overlapped with 380 genes related to small molecule catabolic processes, lipid catabolic processes, tricarboxylic acid cycle, regulation of fibrinolysis, negative regulation of coagulation, and fatty acid beta-oxidation. The testis had 168 functional sites related to hypothalamus development and oxidoreductase activity, acting on the CH-CH group of donors in 142 genes. Similarly, the duodenum had 246 genes (290 sites) enriched in the pyruvate metabolic process, glycolytic process, ATP generation from ADP, ADP metabolic process, and microvillus organization, while the jejunum had 331 genes (407 sites) enriched in an intracellular receptor signaling pathway, regulation of GTPase activity, positive regulation of MAPK cascade, immune response-regulating signaling pathway, and positive regulation of insulin receptor signaling pathway. The rectum had 310 genes (383 sites) enriched in macroautophagy, positive regulation of the catabolic process, the establishment of cell polarity, and regulation of ubiquitin-dependent protein catabolic processes (Appendix A). Overall, these results suggest that A-to-I editing is involved in the functional regulation of pig tissues after long-term high-energy diet induction.

#### 2.4.3. Two Functional A-to-I Editing Sites Shared Nine Tissues

There were three A-to-I editing sites shared by those six gut segments, and two of them were also shared by the other three tissues abdominal aorta, liver, and testis. This one site of them (7:24969309-) is located in a transcript (201) of the gene *Swine leukocyte antigen class II, DQ Beta 1 Chain* (*SLA-DQB1*), which are functionally involved in infectious disease, responses to vaccines, and production performance [30] (Appendix A). The other site (17:51259503-) is located in front of the (310 bp) gene *Beta-1,4-Galactosyltransferase 5* (*B4GALT5*), which has immunological regulation roles in porcine reproductive and respiratory syndrome virus (PRRSV) infection [31] (Appendix A).

### 2.5. Characterization of Functional A-to-I Sites in Gene Regions

#### 2.5.1. Functional A-to-I Editing Sites Not Enriched in the Gene Body (Non-Intron)

As motioned before, A-to-I editing plays an important role in pig tissues after causing long-term high-energy intake. To investigate the characteristics of functional A-to-I editing sites in pig tissues after long-term high-energy intake, we performed a fold test using the Fisher Test to estimate their enrichment in gene features and biotypes. We annotated 6041 functional A-to-I editing sites in 2353 genes and found that they were not significantly enriched in any gene biotype, indicating a greater functional potential of A-to-I editing in intergenic regions than in gene regions (Figure 3a and Appendix A). Out of the 2206 protein-coding genes annotated, 82.7% had functional A-to-I sites. Interestingly, the proportion of functional A-to-I editing sites in each gene feature was significantly lower than that of all A-to-I sites (Figure 3b and Appendix A), suggesting that the functional characteristic of A-to-I editing was mainly present in non-gene body regions such as introns or intergenic regions in pigs after long-term high-energy intake, despite the majority of A-to-I editing (87.0%) occurring within genes.

#### 2.5.2. Functional A-to-I Editing Was Not Significantly Related to Alternative Splicing for HED Pig

Transcript alternative splicing involves joining exons in different combinations to generate distinct gene isoforms. Changes in splice sites and flanking sequences can disrupt the recognition of splicing activator or repressor proteins. In this study, we identified A-to-I sites located within 10 bp of the start or end of each exon as splicing-related sites (Figure 3c). Out of the 6041 functional A-to-I editing sites, 364 were splicing-related, and although they were higher than expected, no significant enrichment was found in any type of functional A-to-I site (Figure 3d).

#### 2.5.3. Potential Protein-Coding Change of Functional A-to-I Editing

A-to-I editing occurring in the coding sequences (CDSs) of transcripts can lead to amino acid substitutions. To study the functional characteristics of A-to-I editing involving protein-coding changes, we compared the amino acid sequence before and after editing for 763 A-to-I sites in the CDSs of 1056 proteins. The amino acid composition changed significantly after A-to-I editing, both for all A-to-I sites and for functional A-to-I sites (Figure 3e). Among these sites, 703 (66.6%) resulted in nonsynonymous changes, and 488 (46.2%) resulted in synonymous changes, with 135 proteins (12.8%) having different isoforms depending on the protein they encoded (Figure 3f and Appendix A). However, the amino acid composition varied between HED, Higher and Lower, and ND. We checked for the enrichment of functional A-to-I editing sites in CDSs and found that neither nonsynonymous nor synonymous changes were significantly enriched in this analysis.

#### 2.5.4. Gene Expression-Related Functional A-to-I Editing Enriched in 3′-UTRs

RNA editing has the potential changing gene expression. To investigate A-to-I editing characteristics, we performed the correlation analysis between editing level and gene expression for each A-to-I site and its related protein coding gene (within the gene body). Out of 64,367 A-to-I sites, 20.1% (12,937) were found to have a significant correlation (Pearson *p* < 0.05) with the expression of 1691 genes. The majority of these sites showed a positive correlation coefficient (97.7%, 12,640), with 5.5% (709) having a strong correlation coefficient (absolute values higher than 0.5). The distribution of correlation coefficients can be seen in Figure 3g. Further analysis showed that these A-to-I sites were significantly enriched in the 3′-UTR of protein-coding genes compared to what would be expected based on the background of A-to-I sites in the pig genome (Fisher *p* < 0.05, Figure 3hi). This is interesting because 3′-UTRs frequently contain regulatory regions that play a role in post-transcriptional gene expression regulation. Sequence changes occurring in 3′-UTR regulated mRNA-based processes, including mRNA stability, mRNA localization to subcellular regions, translation, etc. [32]. The enrichment of gene-correlated specific/different editing A-to-I sites between HED and ND pigs over total gene-correlated A-to-I sites suggests that long-term high-energy treatment could potentially influence gene expression by altering the sequence of 3′-UTRs.

Taking together, our analysis of A-to-I editing in pigs indicates that these modifications have a significant potential to regulate gene expression by altering the sequence of 3′-UTRs. Specifically, our findings suggest that A-to-I sites in pigs subjected to long-term high-energy treatment have the greatest potential for regulating gene expression through changes to 3′-UTR sequences, compared to alternative splicing or changes in protein production.

## 3. Discussion

A-to-I editing is a valuable resource in the transcriptome regulation process and plays a critical role in the modification of mRNA stability, alternative splicing, regulator binding, and translation [5,6,8,9,10,11]. It is also well discussed in human diseases [33,34,35,36], while no evidence was found that A-to-I editing is involved in metabolic disorders. To our knowledge, this study is the first systemic investigation of A-to-I editing involved in the development of metabolic disorders.

It has been reported that several factors, including tissular context, genotype, age, feeding conditions, and sex, potentially regulate the editing level of RNA [19,24], while RNA editing controls phenotype via editing the sequence of RNA molecules [16,37]. The A-to-I editing process, which is mediated by the ADAR family, is tissue-specific and enriched in tissue-specific biological functions. Furthermore, the number of RNA editing sites showed a dynamic trend during skeletal muscle development, and the correlation between the editing levels of these sites and their gene expression has been identified [15,16,21]. In addition, studies in cattle have shown that edited genes are significantly related to the development of corresponding tissues [15]. In ovine, conserved A-to-I editing has been found to play a part in the function of genes, as observed through the comparison of kidney and spleen editing [18].

In previous works, we successfully established a long-term high-energy diet-induced metabolic disorder mini pig model, by excessive energy diet of 10% fat and 3% cholesterol for 64 months started on pigs weighing 9.0 to 11.0 kg [3]. The metabolic disorder pig model analyzed metabolic parameters, atherosclerotic lesions, gut microbiome, and host transcriptome, characterized by weight gain, increased serum lipid and proinflammatory cytokine levels, accumulated lipid droplets, higher rate of hepatocytes apoptosis, impairment of intestinal epithelium, and abdominal aortic plaque [3,4]. All of the work is time-consuming, laborious, and costly.

In this study, we used the software REDItools for A-to-I editing identification based on strand-specific, rRNA-free, and short reads data, which is an appropriate solution. Compared with ployA library short reads data, strand-specific and rRNA-free sequencing can not only capture more valuable sequences and give more accurate estimates of transcript expressions but also can localize the sequence reads to a specific strand [38,39]. The distinction of sequence from sense or antisense strand is important for RNA editing identification. When both strands transcribe occurrence, RNA editing can be localized into one strand and give distinctly a signal that how many reads were transcribed by this strand and how many A changed to G at this strand. However, in non-stranded RNA-seq, it can only report the total reads of both strands transcribed and underestimates the proportion of AG changes. When RNA editing occurs in the antisense strand, the sequence signal reports TC changes by non-stranded RNA-seq, but it also can indicate TC changes in the sense strand. So, the strand-specific sequencing strategy technically avoids the confusing base change. Recently, Oxford nanopore technologies developed a technique for A-to-I editing identification that directly identifies inosines in native transcriptomes with high accuracy through convolutional neural network models aggregating signals across multiple reads [40]. The long-read electrical signal for native RNA sequencing technologies has immense potential in the RNA modification area [41,42]. When its cost decreases, we expect it can be applied to non-model animals.

Our analysis shows the highest enrichment of A-to-I editing sites in 3′-UTRs of each gene feature, which is consistent with the previous report [11]. The functional A-to-I sites detected differences or specificity between high-energy induced and control pigs are not significantly enriched in any of gene features, gene biotypes, exon boundary potentially involving alternative splicing, or coding sequence of the transcript with potential to change protein products when comparing with total A-to-I sites in that of regions. This result indicates the functional targets of HED-related A-to-I editing are potentially in preference of non-gene body regions, such as intergenic, intron, etc., although the splicing-related A-to-I sites are higher than expected with no significance (Figure 3). The 3′-UTRs frequently contain regulatory regions post-transcriptionally influencing gene expression. Sequence changes led by A-to-I editing occurring in 3′-UTR can regulate mRNA-based processes, including mRNA stability, mRNA localization to subcellular regions, translation, etc. [32]. Interestingly, the proportion of HED-related gene-correlated A-to-I editing sites in transcript’s 3′-UTRs is 1.7 times for A-to-I sites correlated with expression of its located gene. It suggests more A-to-I sites involved in the base change of transcript’s 3′-UTRs regulating gene expression of pigs under a long-term high-energy inducing, and A-to-I editing in transcript’s 3′-UTRs has the most efforts for response to metabolic disorders.

No significant enrichment of HED-related A-to-I editing sites was found in the non-gene body region, and this analysis focused on gene region has limits. In addition, our strand-specific, rRNA-free, and short reads strategy has disadvantages in A-to-I editing detection of non-coding small RNA, transfer RNA, etc. Oxford nanopore technologies could also be applied in the next step.

## 4. Materials and Methods

### 4.1. Data and Samples

We retrieved data from the SRA database with accession No: PRJEB48889 and PRJNA833901 in our previous works [3,4]. Briefly, these data contains a total of 72 samples from 9 tissues of 8 male inbred Wuzhishan minipigs in this analysis. We selected 6 gut segments, including the duodenum (Du), jejunum (Je), ileum (Il), cecum (Ce), colon (Co), and rectum (Re), and 3 tissues from important organs, including the abdominal aorta (AA), liver (Li), and testis (Te) (Appendix A). They were sequenced by strand-specific, and rRNA-free Illumina HiSeq2500 platform generating 2 × 150-bp paired-end reads. Half of them were randomly selected and treated with a long-term high-energy diet (HED) with 10% fat, 3% cholesterol, and 87% base material. The base material consisted of 20% wheat, 48% corn, 15% soybean cake, 5% fish meal, and 12% rice bran was used to treat the other 4 individuals with a normal chow diet (ND) for 64 months starting with pigs aged 3 to 4 months and weighing 9.0 to 11.0 kg.

To reduce the A-to-I editing detecting bias from the genome variations, we re-sequenced the genome DNA for those 8 individuals extracted from muscle tissue by DNA extraction kit (Tiangen, Beijing, China). Then, a library with an average length of 80 inserts fragments of 350 bp was constructed for sequencing (2 × 150 bp) using a 10X Illumina HiSeq platform sequencing instrument (Illumina, San Diego, USA).

### 4.2. Data Preprocessing

Paired-end (PE) reads of DNA and RNA raw data for each sample were trimmed by Trimmomatic (v 0.39). Using BWA (v0.7.5a) *mem,* we aligned the DNA clean reads to the pig reference genome Sscrofa11.1 (ftp://ftp.ensembl.org/pub/release-107/fasta/sus_scrofa/dna/Sus_scrofa.Sscrofa11.1.dna.toplevel.fa.gz, accessed on 18 January 2023). In addition, RNA clean reads were mapped to the same reference genome by STAR v2.7.9a with the default parameter. The reference index of Sscrofa11.1 for STAR was based on the gene annotation file from the ENSEMBL database (http://ftp.ensembl.org/pub/release-107/gtf/sus_scrofa/Sus_scrofa.Sscrofa11.1.107.gtf.gz, accessed on 18 January 2023). After alignment, both DNA and RNA alignment BAMs were sorted by genome coordinate using Samtools (v 0.1.18). To avoid potential PCR or sequencing optical artifacts influencing, duplicated reads were removed by the MarkDuplicates in GATK v4.1.8.0. Function FixMateInformation in GATK was further used to ensure all information-matching sequencing read paragraph was synchronized with each other.

### 4.3. Variation Calling

BaseCalbrator and ApplyBQSR in GATK were both used to combine single nucleotide polymorphism (SNP) information (http://ftp.ensembl.org/pub/release-107/variation/vcf/sus_scrofa/sus_scrofa.vcf.gz, accessed on 18 January 2023) for sequence alignment file to avoid the random systematic error by recalibration of base quality score. We then used HaplotypeCaller for variation calling, SelectVariants for separation of SNP, and short fragment insertion and deletion (INDEL). No filter process was performed in the variation call to allow the strictest control of subsequent A-to-I editing identification.

### 4.4. Identification of RNA Editing Sites

Based on the sorted and dup-marked BAMs, we used REDItools (v2 on 22 July 2021) for A-to-I editing identification by a strategy of comparative genomic DNA and transcriptome RNA sequence [43]. Firstly, we obtained the RNA mismatches by reditools.py using RNA-seq data with parameter “-s 2 -S -q 25 -bq 25”, which set the strict mode (only sites with edits will be included in the output), first strand oriented for strand-specific RNA-seq data, the minimum read quality as 25, and the minimum base quality as 25. Using the same parameter, we then detect those positions of RNA output for each sample’s DNA data. The script annotate_with_DNA.py was used to match genome location between DNA and RNA sequencing data. Finally, we used selectPositions.py to extract editing sites with parameters “-d 12 -c 5 -C 5 -v 5 -V 5 -f 0.0 -F 0.95 -e -r -u”, set the 12th column as base distribution for DNA-Seq, coverage of RNA-Seq and DNA-seq to 5, bases supporting RNA-Seq and DNA-seq variation to 5, frequency of variation in RNA-Seq to 0, frequency of non-variation in DNA-Seq to 0.95, exclude multiple substitutions in RNA-Seq, exclude invariant sites in RNA-Seq, and use only positions supported by DNA-Seq.

### 4.5. Filtration of RNA Editing Sites

We filtered those candidate editing sites by four steps, including (1) excluding RESs overlapped with known SNPs downloaded from the Ensembl database (release 107) by intersectBed function in bedtools (v2.30.0); (2) removing candidate sites overlapped with individual genome variations SNPs and INDELs that motioned in previous; (3) keeping candidate sites detected in more than (include) 3 individual for at least one tissue; and (4) deleting those sites with editing level equal 100% in all detected samples. Sex chromosomes and unplaced sequences were not involved in this analysis.

### 4.6. Gene Annotation and Functional Enrichment

We intersected RNA editing sites to gene feature regions, including exons, introns, 5′-UTR, 3′-UTR, etc., and intergenic regions using the find overlaps function of the R package GenomicRanges (v1.46.1). The gene annotation of the *Sus scrofa* genome was downloaded from Ensembl (release 107). Gene ontology (GO) enrichment analyses were performed for gene lists of interest based on a human database of org.Hs.eg.db v 3.14.0 as the poorly annotated in pig, respectively, using the enrichGO function from the R (v4.1.2) package clusterProfiler (v4.2.2). All *p*-values were adjusted by the false discovery rate (FDR) method with a threshold of 0.05. xx Protein-protein interaction (PPI) network analysis was used STRING (string-db.org, accessed on 28 January 2023) based on human species annotations.

### 4.7. Principal Component Analysis (PCA)

Based on the matrix generated by A-to-I editing levels of each site for each sample, we carried out principal component analysis (PCA) using the prcomp function from R package stats (v4.1.2). The explained variance of the first principal component (PC1) or second principal component (PC2) was calculated as the standard deviation of PC1 or PC2 divided by the sum standard deviation of the total principal components.

### 4.8. Differential Expression Genes/Transcripts between Two Treatments

Using stringtie v2.1.7, we calculated the read count based on an alignment file and a gene coordinate annotation file (release 107). This step generated a gene count matrix and a transcript count matrix. Gene expression was calculated by normalized read count for full gene length with (The reads Per Kilobase Million) method, which was calculated by the rpkm function from R package edgeR (v3.36.0). Transcript expression also used the rpkm function and was normalized by the total exon’s length of the analyzed transcript. Employing the exactTest function in edgeR (v3.36.0), we detected differential expression genes and transcripts with a significant threshold to *p*-values (FDR) lower than 0.05 and log2 fold change (log2FC) higher than 1.

### 4.9. Differential Editing Sites between Two Treatments

To evaluate the differential editing sites between HED and ND pigs, we compare the RNA editing profiling for those sites detected in at least 3 samples for each HED or ND treatment for each tissue. Differential editing site detection used the Wilcox Test method, and the threshold was set to 0.05.

### 4.10. Definitions of Treatment-Specific Editing Sites

In the HED and ND comparison, we defined treatment-specific editing sites in tissue as those events were only detected in one treatment for at least 3 samples, and other treatment samples were not detected. For example, if an editing site was detected in at least 3 pigs of HED but not in any ND pig, we labeled it a HED-specific editing site.

### 4.11. Correlation between Gene/Transcript Expression and RNA Editing Level

We performed the correlation analysis between RNA editing level and its editing-related gene/transcript expression of each RESs for all samples, using the Pearson method by cor function of the R base package. The threshold was set to *p*-value < 0.05. The absolute values of correlation coefficient of them higher than 0.5 (include) were defined as strong correlation. The RNA editing genes, including *ADAR*, *ARARB1*, and *ADARB2,* were deemed the main factor of RNA editing occurrence. We analyzed their expression and their transcript expression for each tissue and performed the correlation analysis between them and each editing site in the same way.

### 4.12. Protein Coding Change of A-to-I Editing Sites

We focused on these A-to-I editing sites in CDSs, counted the distance range from the start codon to the site’s position of the pig genome, and deleted the intron regions in this range. We then divided the length of obtained coding region by 3, obtaining the position of the counted codon. The edited amino was coded by codon changed A to G on this position. For those A-to-I editing sites located in the negative strand, we converted them to antisense and complementary sequence.

### 4.13. Plotting Gene Regions

Gene features were retrieved from the Ensembl Genome Browser (https://www.ensembl.org/Sus_scrofa/Location, accessed on 24 January 2023). Chromatin states derived using Chromatin Immunoprecipitation Sequencing (ChIP-seq), Assay for Transposase Accessible Chromatin with high-throughput Sequencing (ATAC-seq), high-throughput chromosome conformation capture sequencing (Hi-C), and RNA-seq data of pig were illustrated through the UCSC Genome Browser [44].

## 5. Conclusions

In summary, our study has provided a functional characterization of the transcriptome differences in pig tissues under a long-term high-energy diet, specifically at the RNA modification level. Our findings provide valuable insights into the role of A-to-I editing sites in the development of metabolic disorders, which could have important implications for improving pig health and developing human disease models.

## Figures and Tables

**Figure 1 ijms-24-07921-f001:**
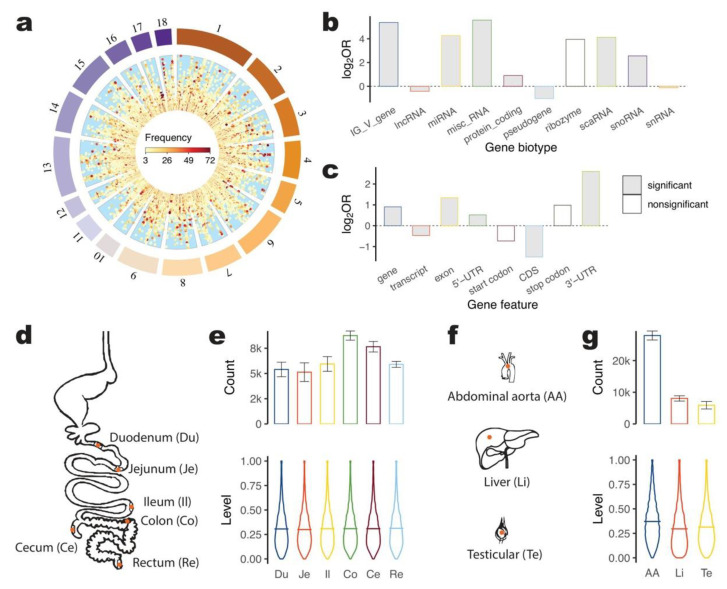
A-to-I editing across pig tissues. (**a**). Circus plot of A-to-I editing genome position. Frequency denotes the count of samples that detected editing sites. (**b**). Fold enrichment of A-to-I editing in each gene biology. Bar plot filling by grey denotes the significant difference detected by Chi-squared Test with *p* < 0.05. Log_2_OR (log_2_ odd ratio) is a statistic that quantifies the strength of the association and is translated by log_2_. (**c**). Fold enrichment of A-to-I editing in each gene feature. (**d**). Diagram of 6 gut segments. Red dots denote the sampling positions. (**e**). Count of A-to-I editing sites and editing level distribution in 6 gut segments. Error bars of bar plots denote the mean count minus/plus standard deviations of samples in each gut segment. At the bottom, the lines denote the mean editing level for each gut segment. (**f**). Diagram of 3 tissues. (**g**). Count of A-to-I editing sites and editing level distribution in 3 tissues.

**Figure 2 ijms-24-07921-f002:**
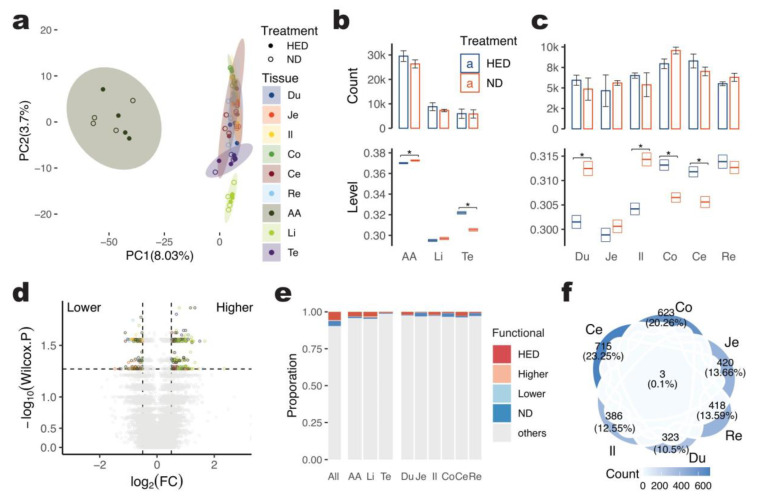
A-to-I editing in the long-term high-energy diet of pigs. (**a**). Principal component analysis (PCA) plot of 72 samples. PC1 denotes the first principal component and follows the explained variance of PC1. (**b**). Count of A-to-I editing sites and editing level distribution of 3 tissues for HED and ND pigs. The star marker * denotes a significant level *p* < 0.05, calculated by Wilcox Test. (**c**). Count of A-to-I editing sites and editing level distribution of 6 gut segments for HED and ND pigs. (**d**). Volcano plot for differently editing A-to-I sites. Grey dots denote no significant sites. (**e**). Proportion of functional A-to-I sites across tissues. (**f**). Venn plot of functional A-to-I sites for 6 gut segments.

**Figure 3 ijms-24-07921-f003:**
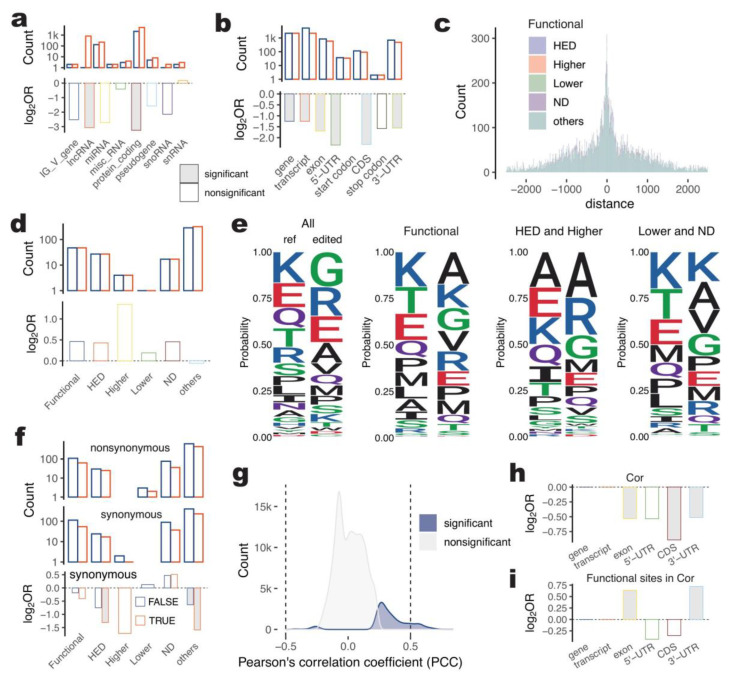
Characterization of functional A-to-I sites in gene regions. (**a**). Count and fold enrichment of functional A-to-I editing sites in each gene biotype. Bar plot filling by grey denotes the significant difference detected by Fisher Test with *p* < 0.05. (**b**). Count and fold enrichment of functional A-to-I editing sites in each gene feature. (**c**). Count distribution of distance for A-to-I editing sites to exon boundary. (**d**). Count and fold enrichment of functional A-to-I editing sites with a distance least than 10 bp to exon boundary in each gene feature. (**e**). The encoded amino acids changed after A-to-I editing. Ref indicates reference amino acid (unedited), edited indicates A-to-I edited amino acid. (**f**). Count and fold enrichment of synonymous or nonsynonymous mutation led by functional A-to-I editing. (**g**). Density plot of gene expression-correlated A-to-I editing sites. Dotted lines denote the threshold of strong correlation. (**h**). Fold enrichment of gene expression-correlated A-to-I editing sites in each gene feature. (**i**). Fold enrichment of gene expression-correlated functional A-to-I editing sites in each gene feature.

**Table 1 ijms-24-07921-t001:** Count of non-redundant A-to-I editing sites for HED and ND treatment of each tissue of pigs.

Treatment	Du	Je	Il	Ce	Co	Re	AA	Li	Te
HED	11,581	12,122	12,907	17,424	15,651	12,579	46,251	15,775	14,359
ND	12,171	11,025	12,610	15,161	17,843	13,148	44,508	13,206	13,457

**Table 2 ijms-24-07921-t002:** Count of HED-related functional editing A-to-I sites for tissues of pigs.

Tissue	Spe/Dif	HED	Higher	Lower	ND
All	6041	3550	278	155	2191
Du	382	311	5	9	57
Je	486	60	7	25	394
Il	456	342	12	26	76
Ce	781	553	13	7	208
Co	698	161	20	21	496
Re	495	137	9	17	332
AA	2027	1423	77	46	481
Li	823	537	111	8	167
Te	231	120	28	-	83

## Data Availability

The data supporting this research’s results are available in the article and Appendix A.

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
