# Peer review of "Characterization of A-to-I Editing in Pigs under a Long-Term High-Energy Diet"

_ijms, 2023, doi:10.3390/ijms24097921_

Round 1
Reviewer 1 Report
In the study, the authors examined the characterization of A-to-I editing in pigs under a long-term high-energy diet. The subject of this study is suitable for the “Animals” journal. The authors indicated that A-to-I editing has the most significant potential in regulating gene expression by changing the 3’-UTRs’ sequence, which is functionally involved in pigs under a long-term high-energy diet. The manuscript's topic is interesting and relevant in the field, and the authors provided new information regarding sow nutrition for pig farming. The study has been well presented. The design and management were well described. The results and discussion are well. The references are appropriate. Tables and figures are given clearly and understandably. I suggest a few corrections to increase the scientific value of the manuscript. After the authors address these corrections, the manuscript can be accepted.
1. A-to-I RNA editing is the most common regarding the range of organisms affected, the breadth of tissues edited, and the number of editing sites. Therefore, the aim of the study concerning a long-term high-energy diet with A-to-I RNA editing in pigs is not precise. An aim to express more descriptive work should be added to the abstract and the introduction section.
2. Metabolic abnormalities caused by long-term feeding with a high-energy diet in pigs, the solutions applied, if any, and the possible effects on the pig industry should be briefly mentioned in the introduction.
3. The results have been well presented, and the discussion has supported the results, but the conclusion may be improved.
4. It needs to be clarified how long the pigs were fed in the study and how many pigs were fed; the control group is not fully understood. This part should be rewritten more understandably.
Author Response
We are grateful to the editor and reviewers for their valuable comments and suggestions on our manuscript. We have carefully considered all of the critiques and have made the necessary revisions to improve the quality of our work.
In the study, the authors examined the characterization of A-to-I editing in pigs under a long-term high-energy diet. The subject of this study is suitable for the “Animals” journal. The authors indicated that A-to-I editing has the most significant potential in regulating gene expression by changing the 3’-UTRs’ sequence, which is functionally involved in pigs under a long-term high-energy diet. The manuscript's topic is interesting and relevant in the field, and the authors provided new information regarding sow nutrition for pig farming. The study has been well presented. The design and management were well described. The results and discussion are well. The references are appropriate. Tables and figures are given clearly and understandably. I suggest a few corrections to increase the scientific value of the manuscript. After the authors address these corrections, the manuscript can be accepted.
- A-to-I RNA editing is the most common regarding the range of organisms affected, the breadth of tissues edited, and the number of editing sites. Therefore, the aim of the study concerning a long-term high-energy diet with A-to-I RNA editing in pigs is not precise. An aim to express more descriptive work should be added to the abstract and the introduction section.
>> Revised. Description of RNA editing is supplemented in abstract, and previous work constructed pig A-to-I editing profile is descripted in the introduction. See in page 1, line 14-15 “Long-term high-energy intake has detrimental effects on pig health and elevates the risk of metabolic disease. RNA editing modifying RNA base in a post-transcriptional process has been extensively studied for model animals.”; and page 2, line 60-65 “Several factors, such as temperature, tissue type, genotype, diet, and age, can influence RNA editing. The study used RNA sequencing data to identify A-to-I editing sites in the pig genome and found that A-to-I editing was widespread in pigs, occurring in over 10,000 genes. The study also found that A-to-I editing levels varied between different tissues and developmental stages, suggesting that A-to-I editing may play tissue-specific roles in pigs.”
- Metabolic abnormalities caused by long-term feeding with a high-energy diet in pigs, the solutions applied, if any, and the possible effects on the pig industry should be briefly mentioned in the introduction.
>> Revised. The possible effect on the pig industry is mentioned in the introduction. See page 2, line 72-76 “Not only that, pigs are a valuable source of protein, as their meat is a staple food item for people. Pig industry is a major contributor to the global economy. Maintaining good pig health is essential for the sustainability and profitability of the pig industry, as well as for ensuring food safety and animal welfare.”
- The results have been well presented, and the discussion has supported the results, but the conclusion may be improved.
>> Revised. We improved the conclusion and rewrote the sentence with “In summary, our study has provided a functional characterization of the transcriptome differences in pig tissues under a long-term high-energy diet, specifically at the RNA modification level. Our findings provide valuable insights into the role of A-to-I editing sites in the development of metabolic disorders, which could have important implications for improving pig health and developing human disease models.” In page 10-11, line 1424-1428.
- It needs to be clarified how long the pigs were fed in the study and how many pigs were fed; the control group is not fully understood. This part should be rewritten more understandably.
>> Revised. We added the information for ingredient of long-term high-energy diet and normal chow diet groups. Please see in page 4, line 399-403 “we treated half of the pig individuals with a long-term high-energy diet (HED, with added 10% fat, 3% cholesterol, and 87% base material) for 64 months starting with pigs aged 3 to 4 months. The nonredundant A-to-I editing sites for HED and normal chow diet (ND, with 20% wheat, 48% corn, 15% soybean cake, 5% fish meal, and 12% rice bran) treatment of each tissue of pigs”.
Reviewer 2 Report
Whilst the work shows an interesting approach and the results are intriguing the manuscript is currently a little hard to follow. I believe that some revision by an English speaking editor would clarify matters. I also think that the discussion might be extended to include some further ideas concerning the location of the changes in editing uncovered and their possible functional significance. It is alluded to but seems to be given as an indication of future work.
Author Response
We are grateful to the editor and reviewers for their valuable comments and suggestions on our manuscript. We have carefully considered all of the critiques and have made the necessary revisions to improve the quality of our work.
Whilst the work shows an interesting approach and the results are intriguing the manuscript is currently a little hard to follow. I believe that some revision by an English-speaking editor would clarify matters. I also think that the discussion might be extended to include some further ideas concerning the location of the changes in editing uncovered and their possible functional significance. It is alluded to but seems to be given as an indication of future work.
>> Revised. We did English editing and rewrote all parts of the results, see page 2, line 86 to page 9, line 1113. We added the functional significance of A-to-I editing in the discussion, see “It has been reported that several factors, including tissular context, genotype, age, feeding conditions, and sex, potentially regulate the editing level of RNA, while RNA editing controls phenotype via editing the sequence of RNA molecules. The A-to-I editing process, which is mediated by the ADAR family, is tissue-specific and enriched in tissue-specific biological functions. Furthermore, the number of RNA editing sites showed a dynamic trend during skeletal muscle development, and the correlation between the editing levels of these sites and their gene expression has been identified. In addition, studies in cattle have shown that edited genes are significantly related to the development of corresponding tissues. In ovine, conserved A-to-I editing has been found to play a part in the function of genes, as observed through the comparison of kidney and spleen editing” on page 10, lines 1358-1368.
Round 2
Reviewer 2 Report
You have worked hard in editing the manuscript and brought the writing into line with the scientific content. An interesting addition to the literature.